# Chronic Nodular Prurigo: An Update on the Pathogenesis and Treatment

**DOI:** 10.3390/ijms232012390

**Published:** 2022-10-16

**Authors:** Lai-San Wong, Yu-Ta Yen

**Affiliations:** 1Department of Dermatology, Kaohsiung Chang Gung Memorial Hospital and Chang Gung University College of Medicine, Kaohsiung 833, Taiwan; 2Institute of Biomedical Sciences, National Sun Yat-Sen University, Kaohsiung 804, Taiwan; 3Department of Dermatology, Fooyin University Hospital, Pingtung 928, Taiwan

**Keywords:** pruritus, chronic nodular prurigo

## Abstract

Chronic nodular prurigo (CNPG) is a recalcitrant chronic itchy disorder that affects the quality of life. It can be triggered by multiple etiologies, such as atopic dermatitis, diabetes, and chronic renal diseases. The mechanisms of CNPG are complicated and involved the interaction of the cutaneous, immune, and nervous systems. Diverse immune cells, including eosinophils, neutrophils, T cells, macrophages, and mast cells infiltrated the lesional skin of CNPG, which initiated the inflammatory cytokines and pruritogens release. In addition, the interaction between the immune cells and activated peripheral sensory nerve fibers by neurotransmitters caused neuroinflammation in the skin and intractable itch. This itch-scratch vicious cycle of CNPG results in disease exacerbation. CNPG is difficult to treat with traditional therapies. Recently, great advances have been made in the pathophysiology of both inflammation and pruritus transmission in CNPG. In this review, we summarize the updated mechanisms and novel therapies for CNPG.

## 1. Introduction

Chronic nodular prurigo (CNPG), also known as prurigo nodularis, is a distinctive chronic itchy disorder [1]. It is a subtype of chronic prurigo (CPG). CPG is defined by pruritus for more than 6 weeks, evidence of chronic scratching, and the presence of multiple pruriginous lesions [2]. According to the clinical phenotype, CPG is classified as five subtypes: papular (papules smaller than 1 cm), plaque (flat plaque > 1 cm), nodular (equal to prurigo nodularis, nodule > 1 cm), linear (linear arrangement), and umbilicated (ulcers within the lesions) types [3]. The different subtypes may coexist [2,4]. Among the subtypes of CPG, CNPG is characterized by intensively pruritic hyperkeratotic and dome-shaped nodules more than one cm [1,5]. A European cross sectional-study revealed that 406 out of 509 patients with CNPG experienced moderate to severe itch intensity scores [6]. In addition to high intensity of itch severity, the pruritus in CNPG is often persistent and associated with burning, stinging, and pain [6,7]. Meanwhile, the lesions have a negative impact on their appearance, daily activities, and quality of sleep [6]. Increased incidence of mental health disorders has been found as well [8]. A meta-analysis study, which included 13 studies, demonstrated a reduction in quality of life in all the studies [9]. It indicates a high burden and negatively affected the quality of life (QOL) in patients with CNPG.

The underlying etiologies of CPG can be heterogenous and multifactorial including dermatological, systemic, neurologic, psychiatric, the combination of the above conditions, and unknown [2]. Psychiatric diseases, diabetes, atopic eczema, and allergy have been reported to be the most common associated diseases with chronic prurigo [6]. Huang et al. showed that CNPG was associated with chronic renal disease, followed by cerebrovascular diseases and type 1 diabetes mellitus [8]. In addition, comorbidities such as pulmonary disease, cancer, human insufficiency virus (HIV), and metastatic disease have been linked to CNPG [10]. Furthermore, an increased likelihood of allergy conditions with CNPG was also found [8]. A consecutive cohort study of 109 patients found that 46.3% of patients with CNPG had an atopic predisposition or atopic dermatitis (AD) [11].

The pathophysiology of CNPG is multifactorial and complicated. Recently, great progress has been made both on the possible pathogenesis and targeted therapies of CNPG. In clinical and physiological settings, this review aims to focus on the pathogenesis (Figure 1) and the updated treatments of CNPG.

## 2. Pathogenesis

It is recognized that the dysfunction of the immune system and immune-neurologic network plays a key role in the pathogenesis of CNPG [12]. Recently, a transcriptomic study demonstrated an upregulation of TGFβ-induced epithelial-to-mesenchymal transition, epidermal acanthosis, axon regeneration, and vascular endothelial growth factor (VEGF) activity in the lesional skin of CNPG [13], indicating the dysregulation of multiple systems is involved in the pathophysiology of CNPG.

### 2.1. Keratinocytes Signaling

Hyperkeratosis, epidermal hyperproliferation, hypergranulosis, and acanthosis are the distinctive feature in the lesions of CNPG, suggesting keratinocytes take part in the physiopathology of CNPG [14]. However, it is unclear whether alternations of the structure of keratinocytes are the consequence of scratching or executor of CNPG. Abnormal differentiation has been revealed in keratinocytes of patients with CNPG [15]. Mitotic activities were increased with upregulation of keratin 5 (K5)/K14 in basal and suprabasal layers of the epidermis and terminal differential K1 and K10 in the suprabasal layer were increased expression in patients with CNPG [15]. It suggests abnormal keratin expression might contribute to the clinical and histological features of CNPG. On the other hand, keratinocyte is one of the big sources of pruritogens, growth factors, and inflammatory cytokines [16,17,18]. Zhong et al. showed that upregulation of mRNA expression of nerve growth factor (NGF), high-affinity NGF receptor (NGFR), tyrosine kinase receptor A (TrkA), and thymic stromal lymphopoietin receptor, while downregulation of the low-affinity NGF receptor, p75 neurotrophin receptor (NTR), interleukin (IL)-31/ IL-31 receptor A (RA), endothelin (ET)-1, ET-2, ET-3, and ET receptor B (ETBR) in patients with CNPG and nearly half of the patients had an atopic history [19]. It indicates that mediators in keratinocytes contribute to an inflammatory response by coordinating with the immune cells by related receptors [19].

### 2.2. Inflammatory Cytokines and Mediators

A variety of immune cells, including eosinophils, neutrophils, T cells, macrophages, and mast cells have been found infiltrating in lesional skin of CNPG [20,21,22,23,24]. In which, enhanced eosinophil degranulation, increased numbers of mast cells, and a distinct subtype of mast cells, dendritic mast cells, were displayed [22,25]. In addition, diverse inflammatory cytokines are involved in the pathogenesis of CNPG and increased mRNA expression of IL-4, IL-17, IL-22, and IL-31 in the skin lesions of prurigo has been reported [26]. We summarized the inflammatory mediators, which were proposed to participate in the mechanisms of CNPG, and the proportion of AD predisposition in the studies (Appendix A).

#### 2.2.1. Th2 Cytokines

Previous studies showed T helper (Th) 2 cytokine, IL-4, IL-5, IL-10, and IL-13, were increased in the lesional dermis in patients with CNPG, in which patients with an atopic history have been excluded [27,28], indicating the involvement of Th2 cell in the pathogenesis of CNPG. Fukushi et al. reported that Th2 cytokines activated the STAT6 pathway, which is required for Th2 immune response [29], and has been shown to implicate in CNPG [28]. Additionally, a study showed a correlation between the improvement in disease severity and the expression of STAT 6 in CNPG [30]. However, Belzberg et al. found no significant difference in expression for Th2-related genes, including, IL-4, IL-13, and STAT6 in patients with CNPG, who had no atopy predisposition [24].

#### 2.2.2. IL-31

IL-31 is a well-recognized pruritogenic inflammatory cytokine [31], especially in the pathogenesis of AD [32,33]. It is known that Th2 cells are the main source of IL-31, however, mast cells, eosinophils, basophils, and macrophages are an additional source of IL-31 [33]. IL-31 signals via IL-31 receptor α chain (IL-31 RA) and oncostatin M receptor β (OSMRβ) chain [33]. IL-31 receptors are expressed in a range of cells [33] and keratinocyte is one of the major targets [34]. IL-31 can induce epidermal-basal cell proliferation in mice [35], suggesting the possible role of IL-31 on epidermal hyperproliferation in CNPG. It showed that the itch intensity of pruritus was correlated to the number of IL-31+ cells, IL-31RA+ cells, and OSM+ cells in the dermis of CPNG lesions [36]. The above study also demonstrated that the IL-31+ cells were predominantly T cells and macrophages while the dermal IL-31RA+ cells were mast cells and macrophages [36]. However, the ratio of patients with an atopic history were not mentioned in the study. Another group, which recruited 11.6% patients with an atopic history, demonstrated increased serum levels and lesional expression of IL-31 in patients with CNPG, however, there was no correlation with the itch intensity [37].

#### 2.2.3. Th22/IL-22

IL-22 is a member of IL-10 cytokine family and is produced largely from immune cells, including Th17 cells, Th22 cells [38], innate lymphoid cells [39], natural killer (NK) T cells [40], and γδ T cells [41]. IL-22 receptor (IL-22R) is composed of IL-22R1 and IL-22RA2 or IL-10R2 chains, which are expressed in epithelial cells and dermal fibroblasts in the skin [42]. IL-22 is characterized by keratinocytes proliferation via the STAT3 pathway and inhibits keratinocytes differentiation [42,43,44]. It also mediates dermal inflammation in the skin [45,46]. Dysregulation of IL-22 has been found to involve in the pathogenesis of psoriasis [47] and AD [48]. We showed that increased lesional IL-22 expression in immunohistochemistry staining [49]. Furthermore, Belzberg et al. reported a significant expression of Th22/IL22-associated genes, IL-22RA1 and IL22RA2 in lesional skin of CNPG. Increased Vδ2 + γδ T cells in the peripheral blood mononuclear cells (PBMCs) of patients with CNPG and activated T cells showed Th22 polarization. Moreover, the itch severity was strongly correlated with the degree of Th22-related gene dysregulation [24].

#### 2.2.4. Th17/IL-17

IL-17 is a crucial proinflammatory cytokine in various skin dermatoses, such as psoriasis [50]. IL-17 consists of six members (IL-17A-F) and Th17 cells are the major source [51]. In addition to the role of host defense and promotion of inflammation [52], it is well known that IL-17 can accelerate keratinocytes proliferation [53,54] and mediate chronic inflammation [55]. Park et al. revealed significantly higher Th17 cells in prurigo lesions without mentioning the ratio of patients with atopy predisposition [26]. Consistently, our group also demonstrated that significantly increased Th17 cell infiltration in the lesional skin in patients with CNPG, in which 25% of patients with an atopic history, and IL-17/ET-1 participated in the pathogenic pathway [49]. Recently, Belzberg et al. revealed increased Th17/IL-17-induced genes, such as S100 genes (S100A7/A8/A9/A12), LOR, and IL36G, in the lesional skin of CNPG as well [24]. However, no systemic increase in IL-17 was found [24,49]. This indicates the local rather than systemic effect of IL-17 on the pathogenesis of CNPG and the possible regulation of local IL-17 by other immune cells such as mast cells [56] and innate lymphoid cells [57], which are capable of IL-17 release.

#### 2.2.5. ETs

ETs are 21-amino acid peptides and include ET-1, ET-2, and ET-3 isoforms. ETs act through ETAR and ETBR [58]. Among the ETs, ET-1 is a vasoconstrictor and a non-histamine pruritogen, which can induce itch response in both humans [59] and mice [60], and plays a role in inflammatory responses [61] and the pathogenesis of AD [62]. It has been reported that ET-1/ETAR was responsible for pruritus transmission while ETBR had an anti-pruritic effect [63]. On the contrary, the role of ET-2 and ET-3 in pruritus transmission and inflammation is less known [64]. An upregulation of the ET-1/ETAR/ET converting enzyme-1/ERK1/2 axis was revealed in patients with CNPG [59]. In addition, we found the increased circulating and localized lesional ET-1 in patients with CNPG and a pathogenic IL-17/p38/ET-1 pathway in keratinocytes in the pathophysiology of CNPG [49]. However, Zhong et al. reported that decreased immunostaining intensity of ET-3 and ETBR and mRNA expression of ET-1, ET-2, ET3, and ETBR in the lesions of CNPG [19]. Further investigation of the pathological role of ET-1 in CNPG is necessary.

## 3. Neuroinflammation

The skin is densely distributed by peripheral afferent nerve fibers. In which, unmyelinated C fiber and myelinated Aδ fibers are responsible for pruriception. The itch signals from the skin then signal transmit to the central nervous system [65]. Moreover, the primary afferent nerves interact with the cutaneous cells by releasing neuroinflammatory mediators, which can modulate a range of physiological functions, including immunity, inflammation, sensory perception, and cell growth [66]. Dysregulation of the peripheral nerve fibers and neurotransmitters has been shown to contribute to the pathophysiology of inflammatory skin disorders, such as psoriasis and AD [67,68]. Recently, the morphological change of the peripheral nerve fibers [69] and the imbalanced expression of neuropeptides and neuroinflammatory mediators have been shown to have a pathogenic role in CNPG [70].

### 3.1. NGF

NGF is a neurotrophin and is important for the survival and development of the nervous system [71]. As mentioned previously, the cells in the skin are the main sources of NGF, including keratinocytes and mast cells [71]. Furthermore, NGF can promote keratinocyte proliferation reciprocally [72,73]. Upregulated mRNA of NGF and TrkA were found in prurigo lesions [19,74]. Johansson et al. demonstrated a high immunoreactivity of NGF and p75NTR localized in dermal inflammatory cells in lesional skin of CNPG [75]. Furthermore, a previous study showed that NGFR-immunoreactive nerves were localized more densely in areas with more NGF+ cells [75] and another group displayed close vicinity between NGFR+ nerve fibers and histamine-containing mast cells [21], indicating the interaction of immune cells and peripheral nervous system in the skin of CNPG. TrkA expression was significantly enhanced by IL-4 and IL-13 in vitro study of normal human epidermal keratinocytes (NHEK) and overexpression of TrkA can further enhance the proliferation of NHEK, suggesting the interplay between Th2 cytokines and keratinocytes through neuroinflammatory mediator [74]. Furthermore, NGF can sensitize and modulate the peripheral nerves [76], which might contribute to the hypersensitivity to sensory perception [77], the common symptom found in AD [78,79] and CNPG [7].

### 3.2. Neuropeptides

Neuropeptides are released from the cutaneous sensory nerve terminal during inflammation or tissue damage. They are essential mediators for crosslinking the immune and nervous systems [80,81] to modulate itch transmission [82] and immune responses [83]. Substance P (SP) is one of the well-recognized neuropeptides, signaled by the neurokinin 1 receptor (NK1R). In addition to the central and peripheral nervous system, NK1R was expressed in keratinocytes, mast cells, and immune cells [84]. SP is a neuroinflammatory mediator and can induce keratinocytes to release a range of proinflammatory cytokines [85]. The implication of SP and NK1R on pruritus was shown in previous studies [86,87]. Furthermore, SP also binds to Mas-related G-protein-coupled receptors (MRGPRs), which are localized in mast cells and dorsal root ganglion [88,89] and are involved in inflammation and itch signaling [90]. It suggests the SP/MRGPRs pathway is also important for immune response and pruriception. Increased serum levels of SP and epidermal NK1R in patients with CNPG were reported [70]. In addition to SP, Kolkhir et al. reported higher numbers of cortistatin (CST)-expressing cells, CST-expressing mast cells, and MRGPRX2-expressing cells in patients with chronic prurigo [91]. The number of MTGPRX2-expression cells and the serum level of MRGPRX2 correlated with the severity of chronic prurigo. CST, a somatostatin analog, is a neuropeptide and a potent agonist for MRGPR [92]. However, more investigation is necessary for the role of CST in the pathogenesis of CNPG.

### 3.3. Alteration of Peripheral Nerve Fibers

Molina et al. showed neural hyperplasia and enhanced protein gene product 9.5 (PGP 9.5), a pan-neuronal marker, immunoreactive nerve fibers in lesional skin of CNPG but not in lichenification skin of chronic eczema [93], suggesting the specific neuropathic role in CNPG. Increased number and size of peripheral nerve fibers in the dermis of lesional CNPG were reported by other groups [21,94]. Furthermore, the nerve fibers showed a strong expression of p75NTR [94]. In addition to the enhancement in total numbers of peripheral nerve fibers in the dermis, increased expression of peptidergic nerve fibers, SP + and calcitonin-gene related peptides (CGRP) + nerve fibers, were found [93,95,96], reflecting activation of the neuroinflammatory microenvironment in CNPG. Moreover, increased amounts of eosinophils were distributed closely with PGP 9.5 + nerves [20] and histamine-containing mast cells were also in close proximity to the NGFR + nerves [21] in the lesion of CNPG, indicating a high level of interaction of immune cells and peripheral never fibers through inflammatory mediators such as NGF in the lesional skin of CNPG. In contrast to the hyperinnervation in the dermis, small fiber neuropathy (SFN) with a reduction in intraepidermal nerve fiber density (IENFD) was revealed in both lesional and nonlesional skin of CNPG [97], however, Bobko et al. only found a reduction in IENFD in lesional skin but not in nonlesional skin [69]. Unlike other etiologies of SFN [98], Pereira et al. found the diminution of IENFD in lesional skin without changes in the sensory tests and endogenous molecular markers, including nerve elongated factor, NGF, and nerve repulsive factor, semaphorin 3A [99]. Furthermore, reconstitution of the IENF was demonstrated in healed lesions [69]. This indicates scratching might contribute to the mechanical reduction in peripheral nerve fibers.

## 4. Treatment

Firstly, recognition of the underlying diseases and breaking the itch-scratch cycle are essential for CNPG treatment. There is no approved systemic licensed treatment and most of the therapies for CNPG are off-label administration. Guidelines for the treatment of CNPG were published by the Japanese Dermatological Association [100] and the International Society for the Study of Itch [3], respectively. They have provided excellent guidelines for diagnosis and treatment strategies for clinical practice. In brief, topical corticosteroid was effective in the treatment of prurigo with precaution for long-term adverse effects and avoid in treatment-resistant patients. Next, topical vitamin D3 analog, tacrolimus ointment, and topical capsaicin were considered when treatment with topical corticosteroid was insufficient. For systemic treatment, antihistamine is generally the first-line treatment. Ultraviolet light therapy can be used in recalcitrant patients. Next, gabapentin, pregabalin, antidepressant, and immunosuppressant, such as cyclosporine and methotrexate, might be considered. However, treatment of CNPG remains difficult while traditional therapies often showed insufficient responses. There are emerging therapies targeting the pathophysiology of CNPG and have been shown efficacious responses (Table 1).

### 4.1. SP and NK1R

Aprepitant, an NK1R antagonist, showed a significant decrease in visual analog scale (VAS) for pruritus from 6.33 ± 1.3 to 4.46 ± 2.88 after 4-week oral administration of 80 mg/day in a study with 12 patients. Downregulation of ERK1/2 MAPK signaling with decreased expression of inflammatory B and T cell markers, CD5 and CD25 were found after aprepitant treatment, suggesting the peripheral cutaneous role of NK1R in the pathogenesis of CNPG [101]. A trend of reduction in the intensity of pruritus and clinical scores with topical aprepitant without significance was reported [70]. Oral administration of 5 mg serlopitant, an NK1R antagonist, once daily showed a significant improvement in pruritus VAS score at week 4 and week 8 in a phase 2 trial of 128 patients [102]. A study on the long-term safety of serlopitant was terminated due to a corporate decision (NCT03540160). Capsaicin, a protoalkaloid in hot peppers, links to transient receptor potential vanilloid 1 (TRPV1), which is a well-known pruriceptor [113] and leads to a release of neuropeptides once activated. Continuous stimulation of TRPV1 by capsaicin has been shown to neuronal desensitization and depletion of SP from primary sensory neurons [114,115]. A case series of 33 patients with CNPG showed effectiveness in clearing skin lesions with topical capsaicin (the concentration was stepped up from 0.025% to the highest concentration of 0.3%) [103].

### 4.2. Gabapentin/Pregabalin

Gabapentin and pregabalin, γ-aminobutyric acid (GABA) analogs, are ligands of the α2δ subunit of voltage-dependent calcium channels [116]. In addition to the licensed treatment for anticonvulsants, they have been shown to be effective for the clinical use of anxiety [117]. Intolerable itch which leads to the desire to scratch is the characteristic feature of CNPG. Thus, breaking the vicious itch-scratch cycle by cognitive modulating itch and attenuating scratching behavior is one of the important strategies in the treatment of chronic pruritus and CNPG [12,118]. GABA can also downregulate neurotransmitter release and neuronal hyperexcitability, which is one of the possible mechanisms for the elimination of pruritus [119]. It has been found efficacious GABA analogs in the treatment of chronic pruritus in various etiologies, including uremic pruritus, brachioradial pruritus, and pruritus of cholestasis [120]. The efficacy of gabapentin with a final titrated dosage of 1800 mg/day has been shown in the treatment of pruritus of unknown origin in two cases [114]. For CNPG, twenty-three out of thirty patients responded with a significant improvement in VAS scores with good tolerance after one-month treatment of pregabalin at a dose of 75 mg/day [104]. A case report showed a rapid reduction in VAS scores from 10 to 0 within one day at an initial dose of 150 mg/day of pregabalin and a maintenance dose of 225 mg/day [105].

### 4.3. Opioid

Pruritus is a common adverse effect following opioid use and μ opioid receptor was primarily involved in opioid-induced pruritus [121]. A case report showed naltrexone, a μ opioid antagonist, attenuated pruritus in a patient with prurigo excoriate [122]. On the other hand, the κ opioid receptor has been found to attenuate itch [123] and difelikefalin, a κ opioid receptor agonist, demonstrated good efficacy and has been approved for the treatment of uremic pruritus [124,125]. In addition, the imbalances in κ and μ opioid signaling might contribute to the pathogenesis of chronic pruritus [126]. Nalbuphine, a combination of κ opioid agonist and μ opioid antagonist, extended-release tablets showed a significant itch reduction at a dose of 162 mg twice daily in a phase 2 trial with an open-label extension phase study (NCT02174419) [106].

### 4.4. Dupilumab

Dupilumab, a humanized IL-4 α receptor blocker, is approved for the treatment of AD [127]. CNPG is associated with allergy conditions [8] and shared part of pathogenesis with AD, especially type 2 inflammation, suggesting a possible response of dupilumab in the treatment of CNPG. IL-4, the pivotal Th2 cytokine, has been found to potentiate multiple pruritogen responses and the IL-4 α receptor is involved in the pathogenesis of chronic pruritus [128]. Multiple studies have shown the promising efficacy of dupilumab in the treatment of CNPG. Chiricozzi et al. demonstrated that 45.8% (11/24) of patients achieved investigator’s global assessment (IGA) 1 after 16-week therapy. In addition, numeric rating scale (NRS) values for sleeplessness decreased significantly from 8.2 to 1.7 [107]. Significant improvements in health-related QOL scores from 15 to 6 and depression scores from 13 to 6 with three-month dupilumab treatment were reported in a case series of 10 patients [108]. Consistent results were found in patients with CNPG with an atopy history with NRS for pruritus from 10 to 3 after 4-month treatment [109]. A cohort study of 45 patients from eleven articles showed a significant reduction in NRS for pruritus from 8.58 ± 1.89 to 1.78 ± 2.29 after 4-month treatment [129]. Additionally, patients with CNPG initiated later clinical response with dupilumab in comparison with patients with AD and longer treatment was necessary for symptom relief in CNPG patients with AD compared with patients with non-AD [129]. Recently, a retrospective cohort study of 19 patients evaluating the long-term efficacy of dupilumab in generalized chronic prurigo showed that 68.4% of patients achieved IGA 0/1 at 52-week treatment [110]. However, a high proportion of CNPG patients with AD overlap responded to dupilumab than those without AD though there was no statistical significance [110]. More investigation is warranted for the efficacy of dupilumab in CNPG patients without atopy conditions.

### 4.5. Nemolizumab

Nemolizumab, a humanized antibody targeting IL-31RA, has shown efficacy in AD and the associated pruritus [130,131] and is a treatment option for chronic pruritus [132]. A phase 2 clinical trial included 70 patients demonstrated a significant reduction in the peak pruritus scores by 4.5 points in the nemolizumab group (0.5 mg/kg) in comparison with a reduction by 1.7 points in the placebo group at 4-week treatment (NCT03181503) [111]. A post hoc analysis of the above study showed a significant reduction in pruritus within 48 h and ≥4 reductions in peak pruritus NRS was achieved on day 3 [112]. Improvement in scratch and sleep duration was demonstrated by week 1 in nemolizumab group compared to the placebo [112]. Furthermore, a significant decrease in IGA scores and an association of transcriptome changes of suppression of Th2/IL-13 and Th17/IL-17 response were shown after 12 weeks of nemolizumab treatment [13].

### 4.6. Other Emerging Therapies

Janus kinase (JAK) regulates intracellular signaling therefore it can indirectly modulate pruritus transmission by regulating the downstream inflammatory cytokines [133,134]. In addition to the role of inflammation, Oetjen et al. demonstrated that JAK1 in the sensory neurons directly participated in the pathogenesis of chronic pruritus [128]. JAK inhibitors have been shown efficacious in both reductions in the severity and pruritus in AD [135]. A case report showed good clinical response with baricitinib, a JAK 1/2 inhibitor, for the treatment of prurigo-type AD [136]. Tofacitinib, a JAK1/3 inhibitor, with a dose of 5 mg/day to 10 mg/day showed clinical efficacy for CNPG in case reports [137,138]. Abrocitinib, a JAK1 inhibitor, has completed a phase 2 trial for the treatment of CNPG but there is no available data (NCT05038982). A proof-of-concept study of 30 mg twice daily apremilast, a phosphodiesterase (PDE) 4 inhibitor, showed no reduction in pruritus in 7 patients after 12 weeks of treatment [139]. A phase 2 a/b clinical trial of vixalerimab (KPL-716), an OSMRβ, for treatment of CNPG is now ongoing (NCT03816891). Barzolvolimab (CDX-0159), a humanized monoclonal antibody binding to the receptor tyrosine kinase KIT, is now recruited for the phase 1 study (NCT04944862).

## 5. Conclusions

In summary, CNPG is a distressing skin disorder and the mechanisms are multifactorial. The interplay between cutaneous, immune, and nervous systems can lead to deeper insights into the pathogenesis of chronic pruritus. Great efforts have been made in the pathophysiology of CNPG recently. It helps to develop more precise therapies for this recalcitrant itch disorder. Given the complex mechanisms in CNPG, more investigation is necessary to explore targeted medicine in the future.

## Figures and Tables

**Figure 1 ijms-23-12390-f001:**
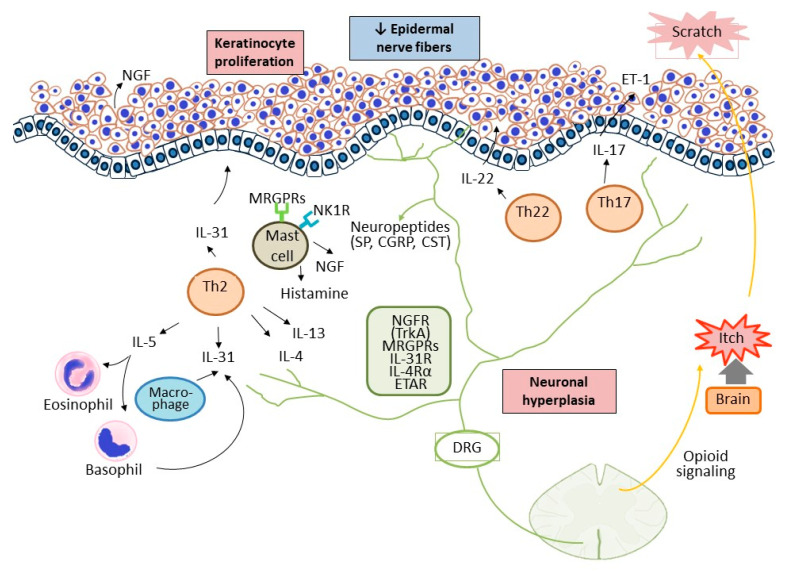
The interplay among the cutaneous, immune, and nervous systems in patients with CNPG. Keratinocyte is a big source of growth factors and inflammatory cytokines, which leads to immune activation. Enhanced infiltration of Th2, Th17/IL-17, Th22/IL-22, eosinophils, and mast cells initiates the inflammatory process and promote keratinocytes hyperproliferation. Simultaneously, neuronal hyperplasia in the dermis further releases neuropeptides, such as substance P, which activate the immune response by interplaying with the immune cells and keratinocytes. The itch-scratch vicious cycle execrates the inflammatory process and scratching causes mechanical damage to peripheral nerve fibers in the epidermis. SP, substance P; NK1R, neurokinin 1 receptor; CST, cortistatin; NGF, nerve growth factor; TrkA, tyrosine kinase receptor A; MRGPRs, Mas-related G-protein-coupled receptors; DRG, dorsal root ganglion.

**Table 1 ijms-23-12390-t001:** Potential targeted treatments in CNPG.

Drug	Target	Possible Mechanisms	Atopic Predisposition
Aprepitant	NK1R antagonist	1. Antagonized the effect of SP 2. ↓ CD5 and CD25 of B and T cells by ↓ ERK1/2 MAPK signaling	83.3% [101]
Serlopitant	NK1R antagonist	Antagonized the effect of SP	43.3% [102]
Capsaicin	TRPV1	Consumption of SP	12.1% [103]
Gabapentin/ pregabalin	α2δ subunit of voltage-dependent calcium channels	Downregulation of neurotransmitter release and neuronal hyperexcitability	No [104] No [105]
Nalbuphine	κ opioid agonist and μ opioid antagonist	Modulation of opioid response	32.2% [106]
Dupilumab	IL-4 α receptor blocker	Downregulation of type 2 inflammation and IL-4 αR-related-itch response	48.1% [107] 44.8% [108] 100% [109] 52.6% [110]
Nemolizumab	IL-31RA	Suppression of Th2/IL-13, Th17/IL-17 and IL-31R related-itch response	11.7% [13,111,112]

CNPG, chronic nodular prurigo; NK1R, neurokinin 1 receptor; SP, substance P; TRPV, transient receptor potential vanilloid; IL-4 αR, IL-4α receptor; IL-31R, IL-31 receptor; Th, T helper. The data above was retrieved by 18 September 2022.

## Data Availability

The data presented in this study are available in the article or the Appendix A.

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
