# Peer review of "Chronic Nodular Prurigo: An Update on the Pathogenesis and Treatment"

_ijms, 2022, doi:10.3390/ijms232012390_

Round 1

Reviewer 1 Report

Major

1. abstract is not informative enough. Please revise it by summarizing the article.

2. The definition of CNPG is vague. Please make it clear whether the study populations in this review all include CNPG related to AD, or systemic diseases such as uremic pruritus, or diabetes in the introduction as well as throughout the article. In particular, CNPG with/without AD should be clearly described in the section of 4.1. Th2 cytokine, 4.2. IL-31, and 4.3. Th22/IL-22.  

- Looking Line 180, CNPG is described as a distinctive disease entity different from AD.

-Studies in Table 1 should clarify whether it includes AD patients as well as the main text of treatment section.

-Line 247: Are these 12 patients AD or CNPG alone?  

3. Fig.1 should include all reviewed substances such as NK1R, CGRP, and TrKA. It would be more informative to show new targeting drugs and their target in the figure.

4. References are not found from 116.

Minor

1. 4.4. Th17/IL-17: Line 134--Please describe IL-17 induced increased genes in detail. Line 137: the possible regulation of IL-17 by other immune cells looks a bit too vague. Please describe more specific.

2. Line 141: 'humansa' should be 'humans'.

3. Line 201:' Alternation  should be 'alteration'.

4. Line 257-260: the participants are all CNPG in ref. 102 & 114?  Please cite the relevant articles only. 

5. What is '0.025%TO0.3%'?

6. Line 319-320:  Only nemolizumab has a sentence about side effects unlike other therapeutic agents. It would be better to remove it. Please cite the reference of NCT03181503 and describe if the patients are CPNG with AD or without AD.

Reviewer 2 Report

Wong et al. summarized the updated pathogenesis of CNPG and the therapies. This review is very well written and very helpful for clinicians treating CNPG. I found very little to criticize.

Minor points

1) Recently, innate lymphoid cells such as ILC2 or LC3 are considered to play an important role in the pathogenesis atopic dermatitis or psoriasis, respectively. Are these cells reported to be involved in the pathogenesis of CNPG?

2) IL-33 is also a Th2-mediated host defense cytokine and considered to be involved in the pathogenesis of AD. Is this cytokine associated with the pathogenesis of CNPG?

3) On page 4 line 160-163, “Recently, the morpho-… role in CNPG.” The authors should provide a reference for this sentence.

4) On page 5 line 198, “CTS” should be “CST”.

Round 2

Reviewer 1 Report

Revision is well done according to the reviewer's comments.

There is some minor issue to need to be corrected.

1. Authors provided Table S1, but it is not reader-friendly. Readers still have to check Table S1 whenever they have questions about patient population. That would reduce readability. Please describe the ratio of AD patients in the main text if it is clearly defined.

2. Line 28: "popular" should be "papular".

3. Line 503: "33 patients" should be " 33 patients with prurigo nodularis".
